# Effect of Boron Doping on Diamond Film and Electrochemical Properties of BDD According to Thickness and Morphology

**Chang Weon Song [1], Dae Seung Cho [2], Jae Myung Lee [1] and Pung Keun Song [3,*]**

[1] Hydrogen Ship Technology Center, Pusan National University, Busan 46241, Korea; cwsong@pusan.ac.kr (C.W.S.); jaemlee@pusan.ac.kr (J.M.L.)
[2] Department of Naval Architecture and Ocean Engineering, Pusan National University, Busan 46241, Korea; daecho@pusan.ac.kr
[3] Department of Materials Science and Engineering, Pusan National University, Busan 46241, Korea
[*] Correspondence: pksong@pusan.ac.kr; Tel.: +82-51-510-3579

**Abstract:** Diamond coating using hot-filament chemical vapor deposition (HFCVD) is now widely used in many fields. The quality of the diamond film and many factors determine the success of the coating, such as temperature, time, and pressure during coating. The purpose of this study was to produce coated boron-doped diamond (BDD) films by doping boron in the diamond film and to assess them through comparative analysis with foreign acid BDD, which is widely used as a water-treatment electrode in the present industry. The bending of the titanium substrate due to the high temperature during the diamond deposition was avoided by adding an intermediate layer with a columnar structure to niobium film. The filament temperature and pressure were determined through preliminary experiments, and BDD films were coated. The BDD film deposition rate was confirmed to be 100 nm/h, and the potential window increased with increasing thickness. The electrochemical activation and catalytic performance were confirmed according to the surface characteristics. Although the high deposition rate of the BDD coating is also an important factor, it was confirmed that conducting coating so that amorphous carbonization does not occur by controlling the temperature during coating can improve the electrochemical properties of BDD film.

**Keywords:** HFCVD; diamond; BDD; electrochemical properties; CV curve

## 1. Introduction

Many studies have been actively conducted to investigate the effects of temperature and pressure changes on diamond coating [1,2]. There are many diamond-coating methods. The most widely used and common of these is hot-filament chemical vapor deposition (HFCVD). Diamond coatings produced by this method have several advantages. It is widely used in many studies because of its application to various substrates. In addition, the biggest advantage of HFCVD is its suitability for industrial use because it can be easily increased in size and can be produced at a low cost compared to other diamond-coating methods [3–5].

Furthermore, research on boron-doped diamond (BDD), which imparts boron to diamond based on excellent diamond properties, gives diamonds electrical properties [6,7]. Therefore many studies have been conducted on BDD because BDD electrodes generate more hydroxyl radicals, more powerful oxidants, on the surface than the most widely used $IrO_2$ electrodes. This results in excellent wastewater treatment [8–11]. Because of these advantages, in the field of insoluble electrodes much research is being conducted on BDD electrodes. However, despite these advantages, BDD is not widely used

industrially because the production cost and price are too high to allow it to be used commercially as an electrode.

The purpose of this study was to manufacture BDD at a lower cost and show better performance than the BDD that is currently available for industrial use. Based on the previous experimental results, the optimal applied power and working distance between the filament and susceptor were determined [12,13]. In order to reduce the manufacturing cost, acetone and trimethyl borate (TMB) were injected into the chamber through a bubbling system and used as carbon and boron sources. In addition, due to its lower cost, we used a titanium substrate rather than niobium, which is commonly used as a substrate for BDD. The electrochemical properties of BDD were analyzed according to their morphology and thickness.

## 2. Experimental Details

Diamond and BDD films were deposited on silicon wafers (100) and titanium plates. The dimensions of the square silicon and titanium substrates were 3 cm × 3 cm, with thicknesses of 500 μm and 1 mm, respectively. A silicon wafer substrate was used to measure the thicknesses of the non-doped diamond and BDD films, and a titanium plate was used to analyze other BDD characteristics. As is well known for silicon wafers, the coefficient of thermal expansion is similar to that of diamond, which facilitates diamond-coating experiments conducted at high temperatures. On the other hand, in the case of diamond coating on a titanium substrate, the thermal expansion coefficients of titanium and diamond differ by 8.6 μm/(mK) and 1.2 μm/(mK), respectively, and the phase transformation from a dense hexagonal lattice structure to a body-centered cubic structure occurs at about 882 °C [14]. Because of this phenomenon, the titanium substrate bends during deposition as shown in Figure 1a) without a niobium interlayer. To solve this problem, physical vapor deposition (PVD) sputtering was used to coat the interlayer of niobium, which is most widely used as a substrate for BDD electrodes, with a columnar structure as shown in Figure 1b) before diamond and BDD deposition. When returned to room temperature after deposition, the columnar structure of the niobium interlayer acts as a buffer between the two materials with different coefficients of thermal expansion. This method, with a niobium interlayer, prevented the titanium substrate from bending as shown on the right in Figure 1a). In addition, preventing the phenomenon of bending can prevent the diamond delamination after deposition. Sputtering was performed through high-power impulse magnetron sputtering (HiPIMs), and the experimental conditions are shown in Table 1. A columnar structure of niobium with a thickness of 3 μm was deposited by 1-hour deposition at 100 °C.

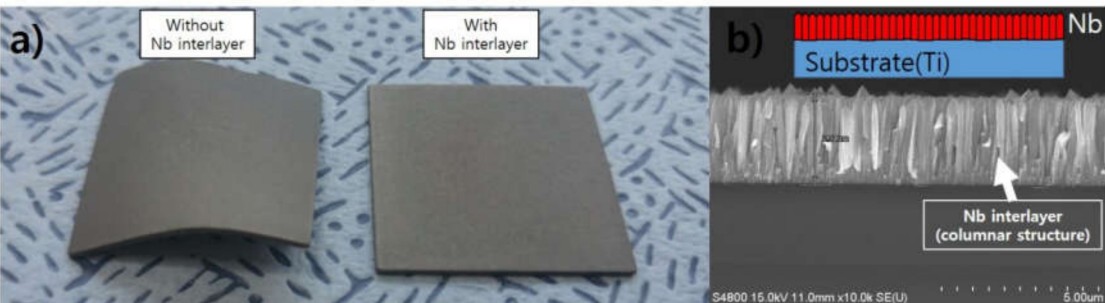

**Figure 1.** (**a**) Diamond deposition on titanium substrates with/without niobium interlayer; (**b**) cross-sectional morphologies of niobium interlayer.

**Table 1.** Details of high-power impulse magnetron sputtering (HiPIMs) experimental conditions.

| Parameter | Detail |
|---|---|
| Base pressure | $5.0e^{-5}$ torr |
| Working pressure | $8.0e^{-3}$ torr |
| Temperature | 100 °C |
| Gas uses | Ar (85 sccm) |
| Deposition Time | 1 h |
| Power source | Nb sputter target 1.0 A |

Figure 2 shows a schematic of HiPIMs equipment used for sputtering of the niobium interlayer and the HFCVD equipment used for deposition of the diamond and BDD coating. Tantalum filaments were used for diamond and BDD coating via HFCVD. The thickness of the tantalum filament was 0.7 mm, the length of each row was 32 cm, and 12 filaments were used. The substrate was deposited at a position 5 cm away from the center of the susceptor in consideration of the low temperature at both ends of the filament. The susceptor was rotated at 1 rpm for uniform deposition. In addition, acetone and trimethyl borate (TMB) were injected into the HFCVD chamber through a bubbling system instead of using methane gas to reduce the cost of BDD coating. At this time, the temperature of the acetone and TMB solution was adjusted using a thermostat. The antifreeze was placed in a thermostat, and the bubbler containing the solutions was immersed in an antifreeze set at 0 °C to bring the solution to 0 °C.

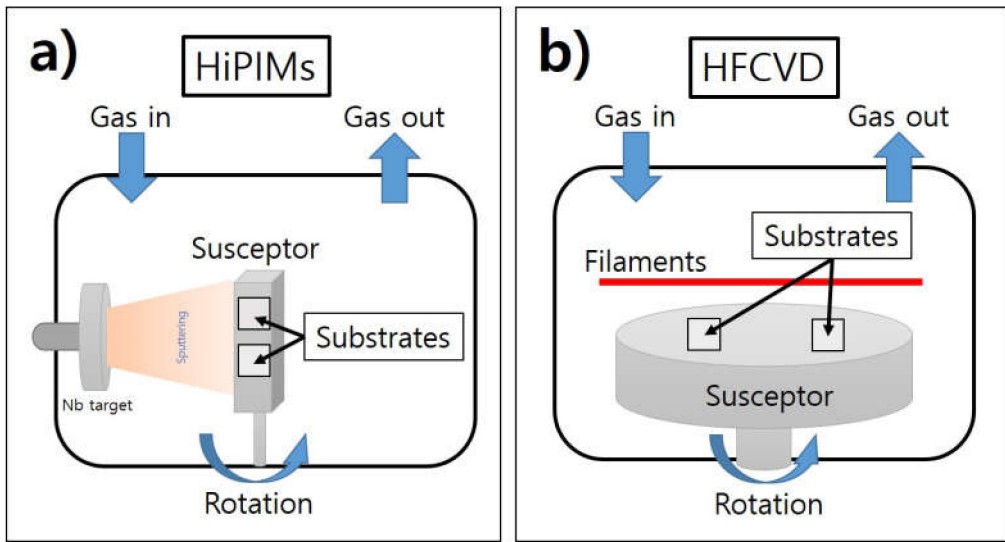

**Figure 2.** Schematic diagram of (**a**) HiPIMs and (**b**) hot-filament chemical vapor deposition (HFCVD).

For diamond coating, pretreatment is necessary [15–17]. If pretreatment is not performed, a diamond film will not be formed even if the experiment is conducted under the same conditions. Diamond powder was used to seed the niobium-coated substrate surface. First, a 500 nm diamond particle powder and glycerin were mixed at a weight ratio of 1:1, and the mixed materials were applied to the substrate surface and rubbed to increase the seeding effect of the diamond particles and increase the roughness of the substrate surface to facilitate diamond coating.

The experimental conditions applied for HFCVD are shown in Table 2. The power applied to the filament, the distance between the filament and the susceptor, and the pressures inside the chamber were 16 kW, 10 mm, and 4000 Pa, which were effective conditions to produce a diamond coating in previous experiments [12,13]. Prior to the BDD coating, the coating of a diamond film not doped with boron was performed for 12 h. This was done to check the morphology and quality of the diamond

film and to compare its morphology with that of the reference BDD. Deposition was carried out for 12 and 60 h for analysis of the morphology, thickness change, and electrochemical properties of BDD film. We use the Antoine equation below to calculate the amount of boron doped in the diamond film:

$$\log_{10} p = A - \frac{B}{C + T} \tag{1}$$

where $p$ is the vapor pressure, $T$ is temperature and A, B and C are component-specific constants. We calculate the vapor pressure of acetone and TMB with A, B and C constant values at 0 °C [18]. The acetone is 70.2047 torr and the TMB is 25.5690 torr. Acetone ($C_3H_6O_6$) and TMB ($C_3H_9O_3B$) was used 90 and 6 sccm, respectively.

$$B/C \text{ ratio} = (\alpha \times 25.5690 \times 6) \div ((\beta \times 70.2047 \times 90) + (\gamma \times 25.5690 \times 6)) \tag{2}$$

where $\alpha$ is the number of boron in TMB, $\beta$ is the number of carbon in acetone and $\gamma$ is the number of carbon in TMB.

$$B/C \text{ ratio} = (1 \times 25.5690 \times 6) \div ((3 \times 70.20474 \times 90) + (3 \times 25.5690 \times 6)) = 0.007902 \tag{3}$$

**Table 2.** Details of HFCVD experimental conditions.

| Parameter | Value |
|---|---|
| Filament power (kW) | 16 |
| Number of filaments | 12 |
| Length of each filament (cm) | 32 |
| Pressure (Pa) | 4000 |
| Distance between filament and susceptor (mm) | 10 |
| Distance between filaments (mm) | 20 |
| Flux of Acetone (sccm) | 90 |
| Flux of trimethyl borate (TMB, sccm) | 6 |
| Flux of Hydrogen (sccm) | 400 |
| Deposition time (h) | 12 and 60 |

Therefore, the diamond film was doped with 7902 ppm of boron.

The morphology and thickness analysis of the diamond and BDD films was performed by field-emission scanning electron microscopy (FE-SEM, S-4800, Hitachi, Tokyo, Japan), and the quality of the diamond film was confirmed by Raman spectroscopy (Horiba, Jobin Yvon, Edison, NJ, USA). Raman spectroscopy was conducted with an argon laser at excitation wavelengths of 514.5 nm and 1800 line/nm gratings. X-ray diffraction (XRD, D8-Advance, Bruker, Billerica, MA, USA) was used for analysis of the BDD film. XRD measurement conditions were measured in coupled scanning diffraction (CSD) mode measuring θ/2θ. The 2θ scan was in the range of 30° to 80°. Cu Kα with the wavelength of λ = 1.5406 Å was used, and the acceleration voltage and current were 40 kV and 40 mA, respectively.

A potentiostat (ZIVE SP2, WonATech Co, Seoul, Korea) was used to analyze the electrochemical properties of the BDD specimens by cyclic voltammetry (CV) curve measurement. The exposed area during the electrochemical test is 10 mm in diameter. The potential window of oxygen and hydrogen reaction was measured using a 0.5 M $Na_2SO_4$ solution as an electrolyte, and the electrochemical activation and catalytic ability of BDD were measured by mixing a 0.5 M $Na_2SO_4$ solution and a 50 mM $K_3Fe(CN)_6/K_4Fe(CN)_6$ solution. Measurement conditions were assessed using a Pt counter electrode and an Ag/AgCl reference electrode to measure the BDD film deposited at a scan rate of 20 mV/s.

## 3. Results and Discussion

The morphologies of the 12 h non-doped diamond and the reference BDD are shown in Figure 3; as seen in the figure, they differ. First, the FE-SEM results in Figure 3a show that the diamond grains are

very uniform in size, and very little amorphous carbonization occurred. On the other hand, in Figure 3b, the blackened part of the BDD can be observed because of the progress of amorphous carbonization due to the high temperature above the optimal temperature during deposition [19–22]. This increase in amorphous carbonization can also explain the decreased deposition rate [12,13]. In terms of average particle size, the non-doped diamond film is smaller than the reference BDD, but triangular (111) facets with superior diamond quality dominate. In Figure 3b, which shows the morphology of the reference BDD, (111) facets and a rectangular shape (100) are observed, and the particle size is also very uneven [23,24]. This has the advantage of increasing the specific surface area of the BDD film, but this results in more diamond etching during deposition, resulting in a reduced diamond deposition rate [25,26].

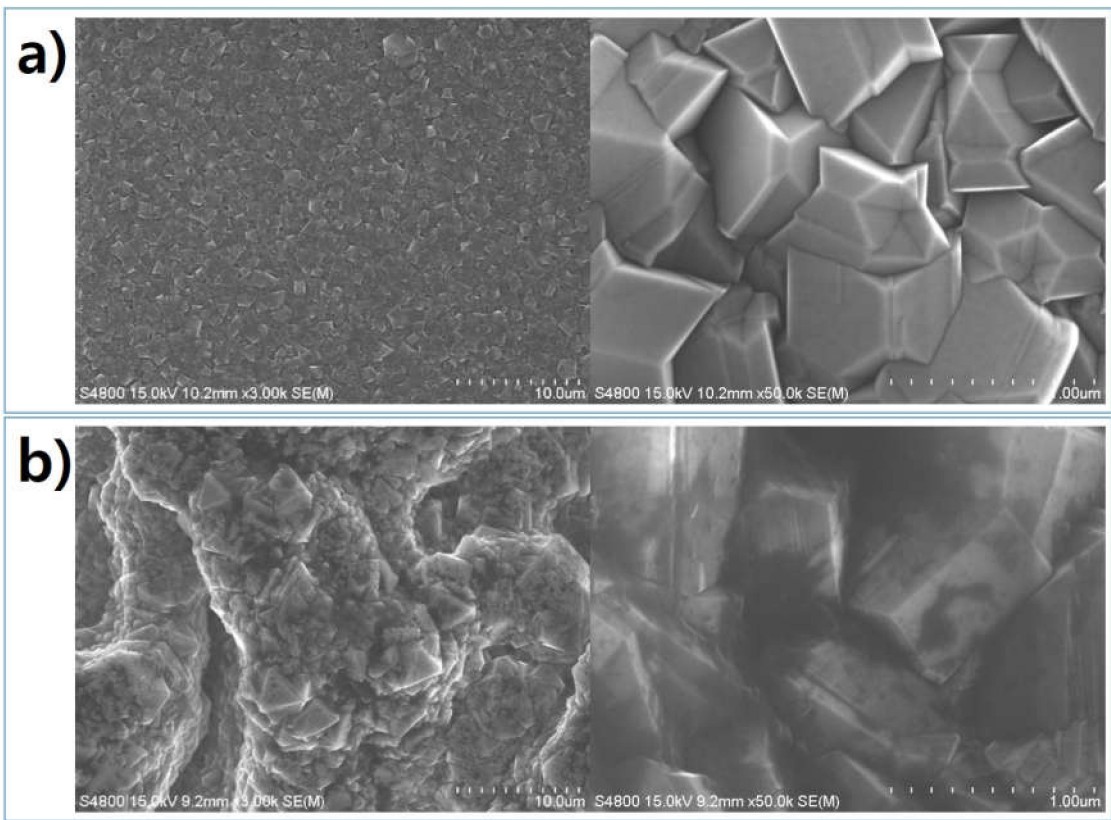

**Figure 3.** Surface morphologies of (**a**) non-doped diamond film and (**b**) reference boron-doped diamond (BDD).

After the diamond film deposition experiment, a boron-doping experiment was carried out. The deposition rate according to the deposition time, changes in the BDD morphology according to boron doping, and changes in the BDD morphology according to the deposition time are shown in Figure 4. First, according to the deposition time, the thickness of the film after 12 h of deposition was 1.22 µm, and the deposition rate was 101.6 nm/h. A thickness of 5.91 µm and a deposition rate of 98.5 nm/h were obtained after 60 h of deposition. It was slightly decreased by about 3% in comparison with the case when deposition was conducted for 12 h. Considering experimental error, it was confirmed that the average deposition rate was 100 nm/h during the BDD deposition. Regarding morphology change according to boron doping, it was confirmed that there was no change in the BDD film morphology due to boron doping in comparison to the diamond film and the morphology shown in Figure 3a). Moreover, regarding the morphology of BDD in relation to deposition time, it can be seen by comparing Figure 4a,b that the particle size and morphology did not change and only the thickness increased with time.

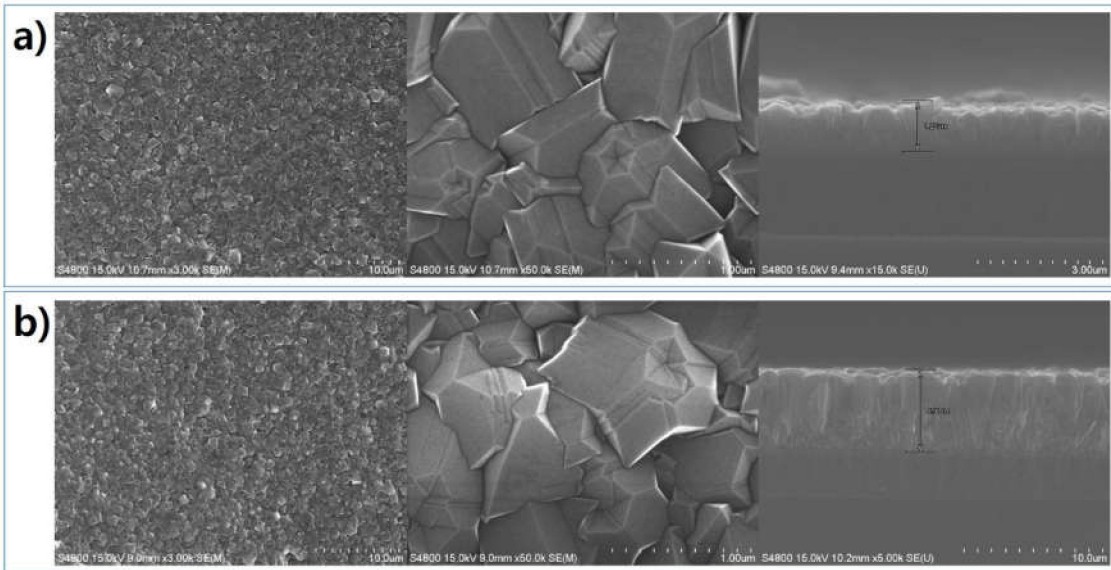

**Figure 4.** Surface and cross-sectional morphologies of the BDD films deposited (**a**) 12 and (**b**) 60 h.

Raman spectroscopy was conducted to analyze the quality of the non-doped diamond, and the results are shown in Figure 5a. A single-crystal diamond (SCD) was used as a reference for comparison with the non-doped diamond. The diamond characterization peak is very distinctly observed at 1332 cm$^{-1}$. In addition, D- and G-mode bands of the amorphous carbon region were observed at 1400–1650 cm$^{-1}$ [27,28]. Raman spectroscopy confirmed that a diamond film was successfully deposited.

Raman analysis showed significant changes as the boron was doped, and there was no difference between the 12 and 60 h samples. Raman spectra of BDD are quite different from those of non-doped diamond films. The corresponding diamond characterization peak at 1332 cm$^{-1}$ in Figure 5b,c are absent. Instead, two broad peaks are observed, shifted to a lower frequency. Boron doping does not significantly affect the FE-SEM morphology of the diamond film, but it does have a large impact on the Raman spectrum.

Regarding this mismatch between the FE-SEM image and the Raman spectra, Ushizawa [23] and Mortet [29] found that when boron doping was higher than 400 ppm, the characteristic peak of the diamond changed. The diamond characterization peak gradually decreases, shifting to a lower frequency, and changes to a broader peak. This would refer to broad peaks found at around 1210 cm$^{-1}$ in Figure 5b,c. This is manifested by impurities and defects as a result of relaxation of wave vector selection rules due to disorders [30–33]. Ushizawa and Mortet also note that when boron is doped with diamond, a broad peak at 490 cm$^{-1}$ is also observed due to interference between continuous electron excitation and the center optic phonon in the discontinuous region [29–33]. Thus, the Raman spectrum in Figure 5b,c indicates that boron was successfully doped into the diamond film.

The XRD pattern analysis results of the BDD films on the titanium substrate with niobium interlayer are shown in Figure 6. Analysis of the patterns was peak matched using joint committee of powder diffraction standards (JCPDS) data. The peaks of niobium carbide can be observed in both the 12- and 60-h BDD film deposition results. Niobium carbide is the peak that appears due to carbonization of interlayer niobium during BDD deposition using HFCVD. The diamond characterization peak in the XRD pattern results at 12 h tends to be lower than that at 60 h due to the increased thickness of the BDD film. Assuming that XRD measures the same area and the same depth, this phenomenon occurs because a deposition time of 60 h includes more BDD in the measurement range than 12 h.

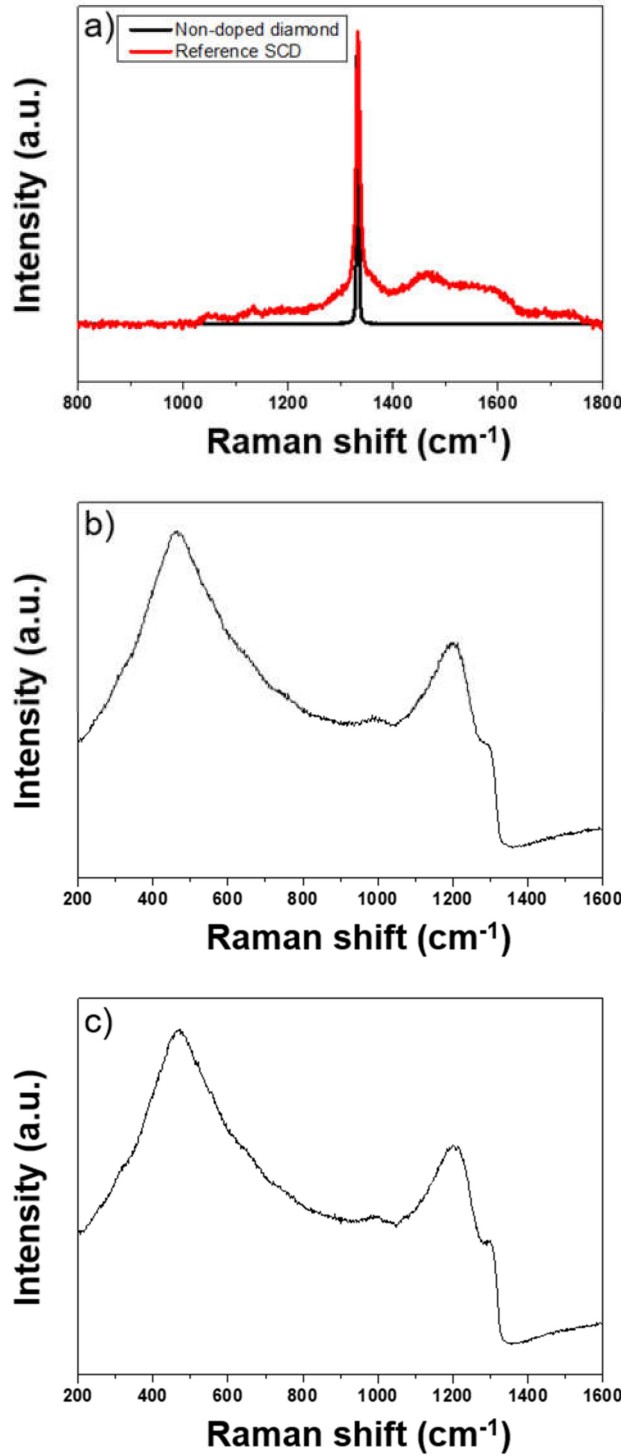

**Figure 5.** Raman spectra of the (**a**) non-doped diamond with reference single-crystal diamond (SCD) and BDD films deposited at (**b**) 12 and (**c**) 60 h.

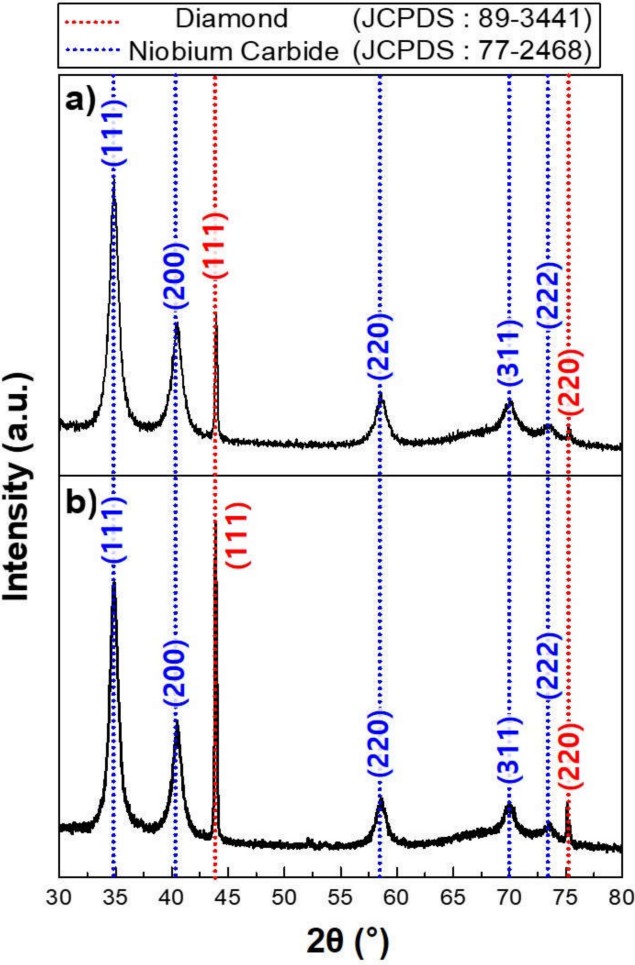

**Figure 6.** X-ray diffraction (XRD) patterns of the BDD films deposited (**a**) 12 and (**b**) 60 h.

Figure 7 shows the results of CV curve measurement using a potentiostat for electrochemical characterization of the BDD film. Figure 7a shows the potential window results. The potential window is the plateau region between the two ends where the inflection point begins. The potential window is the potential range at which the disturbance due to the reaction of electrodes, solvents, and support salts can be ignored for the reaction. In other words, the potential window is influenced by the grain density and thickness of BDD [32,34,35]. It can be seen that the potential window of the deposited BDD film is slightly increased in comparison to that of the reference BDD. In addition, the increase in the potential window with increasing thickness can be confirmed for 12 and 60 h. According to the specification of the reference BDD, the thickness is 10 μm, which is thicker than the thickness of the deposited BDD, but the potential window is reduced due to the reduction of the electrochemical properties caused by amorphous carbonization. In addition to the potential window, in Figure 7b the electrochemical activity and catalytic ability were analyzed to evaluate the water treatment capability and the potential for use as sensors. Catalytic performance and electrochemical activation are not affected by the density and thickness of the BDD film, unlike the potential window, but only by the intrinsic properties of the particles exposed on the BDD surface [32,34,35]. The results for the deposited BDD films were higher than those for the reference BDD film also due to amorphous carbonization. In addition, the results of the deposited BDD films were almost the same because of the intrinsic properties of the particles exposed on the BDD surface with the same ratio of boron doping.

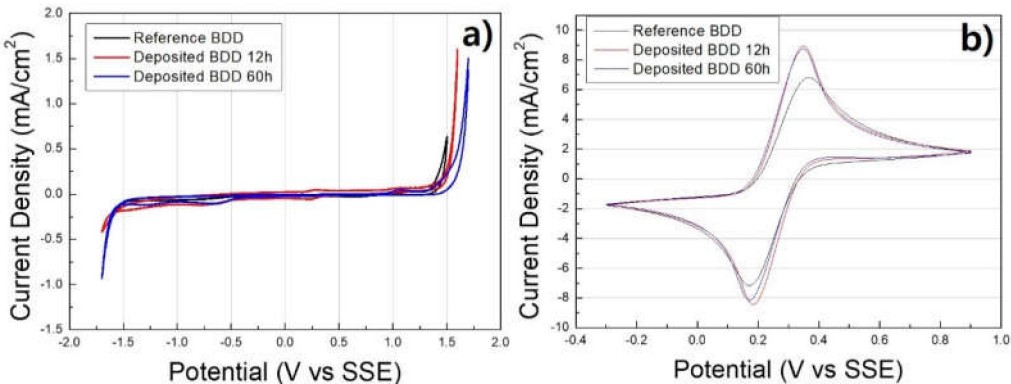

**Figure 7.** Cyclic voltammetry (CV) curves of BDD films in (**a**) 0.5 M $Na_2SO_4$ solution and (**b**) 50 mM $K_3Fe (CN)_6$/$K_4Fe(CN)_6$ solution mixed with 0.5 M $Na_2SO_4$ solution.

## 4. Conclusions

The Raman spectra of the BDD films confirmed that boron was successfully doped into the diamond. However, there was no change in the morphology of diamond film due to boron doping. The experimental results according to the deposition time, the deposition rate according to the increase in deposition time, was confirmed to be constant at an average of 100 nm/h. Furthermore, through the evaluation of electrochemical properties, the BDD film samples had electrochemical properties that the diamond film did not have. As the thickness increased, there was an increase in the potential window, but there was no change in electrochemical activation and catalytic activity. This means that electrochemical activation and catalytic activity are only affected by the properties of the exposed surface, regardless of the thickness or density of the BDD film. Meanwhile, comparison with the reference BDD where amorphous carbonization occurred showed that increased thickness does not necessarily increase electrochemical properties. In other words, although a high deposition rate is an important factor in the deposition of BDD film, deposition under conditions in which amorphous carbonization does not occur due to temperature control during deposition can improve the electrochemical properties of BDD.

**Author Contributions:** Conceptualization, C.W.S. and P.K.S.; methodology, C.W.S.; validation, C.W.S., D.S.C. and J.M.L.; formal analysis, C.W.S.; investigation, C.W.S.; resources, P.K.S.; data curation, C.W.S. and J.M.L.; writing—original draft preparation, C.W.S.; writing—review and editing, P.K.S.; visualization, C.W.S.; supervision, P.K.S.; project administration, P.K.S.; funding acquisition, D.S.C. and P.K.S. All authors have read and agreed to the published version of the manuscript.

**Funding:** This work was partly supported by National Research Foundation of Korea (NRF) grant funded by the Korea government (MSIP) through GCRC-SOP (No, 2011-0030013) and the Technology Development Program of MSS (S2780957) and the Ministry of the Environment (G232019012551) and R&D Platform Establishment of Eco-Friendly Hydrogen Propulsion Ship Program (No. 20006644).

**Acknowledgments:** Our special thanks to Opto-Electronic Materials (OEM) Laboratory members in Pusan National University for assisting in obtaining SEM and XRD data.

**Conflicts of Interest:** The authors declare no conflict of interest.

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
