# Peer review of "Effect of Boron Doping on Diamond Film and Electrochemical Properties of BDD According to Thickness and Morphology"

_coatings, doi:10.3390/coatings10040331_

Round 1

Reviewer 1 Report

1) The article mixes to experiment with theory. It is hard to read and one cannot refer to the elements of current knowledge and research. Part of the introduction is data about the experiment. This should not happen.

2) Research results are shown, but their interpretation is negligible. E.g. Raman spectra. The bands are presented, described, but no conclusions have been drawn from these studies. In many cases, you have a similar impression. This must be definitely improved.

3) The summary is too general and laconic and does not contribute to the publication.

Author Response

We carefully read your comments and tried our best in revising our manuscript according to them. All revisions are highlighted, using the "Track Changes" function in Microsoft Word.

1) The article mixes to experiment with theory. It is hard to read and one cannot refer to the elements of current knowledge and research. Part of the introduction is data about the experiment. This should not happen.

Our reply) We carefully read your comments and make corrections to the overall parts of the manuscript.

And removed the methodology section in the introduction and added some information about the BDD film.

Thank you for your comments.

2) Research results are shown, but their interpretation is negligible. E.g. Raman spectra. The bands are presented, described, but no conclusions have been drawn from these studies. In many cases, you have a similar impression. This must be definitely improved.

Our reply) Renewed interpretation and analysis of the experimental results including Raman. This is included in the results and discussion and conclusion parts.

3) The summary is too general and laconic and does not contribute to the publication.

Our reply) We have revised the manuscript including the conclusion parts.

Reviewer 2 Report

The authors reported the deposition of boron doped diamond films on titanium substrate by HFCVD and investigation of electrochemical properties of BDD film. As the thickness increased, there was an increase in the potential window, but there was no change in electrochemical activation and catalytic activity. The experiments are clearly and explained. The results are logically developed. The work about BDD films would be of interest to the readers. Therefore, I recommend that this work can be accepted after a minor revision.

1, There are many related works about the diamond deposition on Titanium substrates previously. What is the special or novelty of the present study? Which should be stated.

2, The authors use the Antoine equation to calculate the amount of boron doped in the diamond film. The equation should be provided.

3, If the boron was doped into diamond films, the peak located at about 500 cm-1 in Raman spectra can be observed. So, the authors should provide the Raman spectra at about 500 cm-1 to indicate the boron doping.

4, In Figure 6, the diamond JCPDS and niobium carbide JCPDS patterns were not observed. Please added.

5, The author stated that the potential window is affected by the particle density and thickness of the BDD film. What is the meaning of particle density of BDD film?

Author Response

We carefully read your comments and tried our best in revising our manuscript according to them. All revisions are highlighted, using the "Track Changes" function in Microsoft Word.

1, There are many related works about the diamond deposition on Titanium substrates previously. What is the special or novelty of the present study? Which should be stated.

Our reply) If a titanium substrate is used for diamond deposition at high temperatures for a long time, the substrate will bend as shown in figure 1. So we deposited a niobium columnar structure on the titanium substrate to prevent the titanium substrate from bending during diamond deposition.

Additional comments have been added to line 60-66 for clarity.

Before) To solve this problem, PVD sputtering was used to coat the niobium interlayer with a columnar structure as shown in Figure 1b) before diamond and BDD deposition. This method, with a niobium interlayer, prevented the titanium substrate from bending as shown on the right in Figure 1a).

After) To solve this problem, PVD sputtering was used to coat the niobium, which is most widely used as a substrate for BDD electrodes, interlayer with a columnar structure as shown in Figure 1b) before diamond and BDD deposition. When returned to room temperature after deposition, the columnar structure of the niobium interlayer acts as a buffer between the two materials with different coefficients of thermal expansion. This method, with a niobium interlayer, prevented the titanium substrate from bending as shown on the right in Figure 1a). In addition, preventing the phenomenon of bending can prevent the diamond delamination after deposition.

2, The authors use the Antoine equation to calculate the amount of boron doped in the diamond film. The equation should be provided.

Our reply) Thank you very much and sorry. After conducting a literature review based on what you said, we found that the vapor pressure of the TMB was entered incorrectly. Added references to formulas and constant values ​​in manuscript. The Antoine equation and B/C ratio calculation was added to line 100-116 to calculate the amount of boron doped in the diamond film.

Before) Using the Antoine equation to calculate the amount of boron doped in the diamond film, the diamond film was doped with 11400 ppm of boron.

After) Using the Antoine equation below to calculate the amount of boron doped in the diamond film.

Where p is the vapor pressure, T is temperature and A, B and C are component-specific constants. Calculate the vapor pressure of acetone and TMB with A, B and C constant values ​​at 0 ° C [ref]. The acetone is 70.2047 torr and the TMB is 25.5690 torr. Acetone (C3H6O6) and TMB (C3H9O3B) was used 90 sccm and 6 sccm, respectively.

B/C ratio

Where α is the number of boron in TMB, β is the number of carbon in Acetone and γ is the number of carbon in TMB.

B/C ratio

Therefore, the diamond film was doped with 7,902 ppm of boron.

3, If the boron was doped into diamond films, the peak located at about 500 cm-1 in Raman spectra can be observed. So, the authors should provide the Raman spectra at about 500 cm-1 to indicate the boron doping.

Our reply) The current Raman spectra is the diamond before doping boron, not the Raman spectra of boron-doped diamonds. In response to your comments, we have added Raman spectra of boron-doped diamonds.

Due to the addition of the BDD Raman spectra, the original Figure 5 has been modified to Figure 4 and Raman has been replaced with the original 4 to 5 for clarity. There was also the movement and correction of the text due to the change of Figure order.

Before) Raman spectroscopy was conducted to analyze the quality of the deposited diamond, and the results are shown in Figure 4. Single-crystal diamond (SCD) was used as a reference for comparison with the deposited diamond. The diamond characterization peak is very distinctly observed at 1332 cm-1. In addition, D- and G-mode bands of the amorphous carbon region were observed at 1400-1650 cm-1 [25, 26]. Raman spectroscopy confirmed that a diamond film was successfully deposited.

After) Raman spectroscopy was conducted to analyze the quality of the non-doped diamond, and the results are shown in Figure 5a). Single-crystal diamond (SCD) was used as a reference for comparison with the non-doped diamond. The diamond characterization peak is very distinctly observed at 1332 cm-1. In addition, D- and G-mode bands of the amorphous carbon region were observed at 1400-1650 cm-1 [25, 26]. Raman spectroscopy confirmed that a diamond film was successfully deposited.

Raman analysis showed significant changes as the boron was doped, and there was no difference between the 12 hours and 60 hours. Raman spectra of BDD are quite different from those of non-doped diamond films. The corresponding diamond characterization peak at 1332 cm-1 in Figure 5b) and c) are absent. Instead, two broad peaks are observed, shifted to a lower frequency. Boron doping does not significantly affect the FESEM morphology of the diamond film, but it does have a large impact on the Raman spectrum.

Regarding this mismatch between the FESEM image and the Raman spectra, Ushizawa [19] and Mortet [27] found that when boron doping was higher than 400 ppm, the characteristic peak of the diamond changed. The diamond characterization peak gradually decreases, shifting to a lower frequency, and changes to a broader peak. This would refer to broad peaks found at around 1210 cm-1 in Figure 5b) and c). This is manifested by impurities and defects as a result of relaxation of wave vector selection rules due to disorder [19, 28-30]. Ushizawa and Mortet also note that when boron is doped with diamond, a broad peak at 490 cm-1 is also observed due to interference between continuous electron excitation and the center optic phonon in the discontinuous region [19, 27-30]. Thus, the Raman spectrum in Figure 5b) and c) indicates that boron was successfully doped into the diamond film.

4, In Figure 6, the diamond JCPDS and niobium carbide JCPDS patterns were not observed. Please added.

Our reply) The overall modification of Figure 6 has been made. Indexing has been added for each peak. Please check Figure 6 in our revised manuscript.

5, The author stated that the potential window is affected by the particle density and thickness of the BDD film. What is the meaning of particle density of BDD film?

Our reply) This is precisely the density of the BDD film, not the particle density. Sorry for causing confusion. The manuscript has been modified.

Before) Line 202: In other words, the potential window is affected by the particle density and thickness of the BDD film. The potential window is the length from the inflection……

Line 211: Catalytic performance and electrochemical activation are not affected by the particle density and thickness of the BDD film, unlike the potential window, but only by the intrinsic properties…..

After) Line 202: In other words, the potential window is affected by the particle density and thickness of the BDD film. The potential window is the length from the inflection……

Line 211: Catalytic performance and electrochemical activation are not affected by the particle density and thickness of the BDD film, unlike the potential window, but only by the intrinsic properties

Reviewer 3 Report

Introduction is not a place to describe the methodology. Half of this section is devoted to deposition conditions and parameters.

Why Nb interlayer was used and why columnar structure is so important?

What was the wavelength of XRD source ?

What was the size of exposed area during the electrochemical tests?

Line 133 What Authors mean writing “particle size”?

Why Authors did not compare Raman spectra of Diamond and BDD? The visible D and G bands may give a lot of information regarding the amorphous carbon concentration and quality.

Why Authors do not present the XRD spectra of diamond film. Coatings on which substrates were analyzed by XRD?

XRD results can clearly confirm the statement about a dominance of <111> and <100> type crystallographic planes.

Figure 6. First should be presented a) and then b).

The conclusions are not supported by the obtained results. Authors do not refer to any results from other papers. The discussion of the obtained results is at very poor scientific level. Authors rather state than explain the differences, whereas those statements are not supported by proper references.

Author Response

  1. Introduction is not a place to describe the methodology. Half of this section is devoted to deposition conditions and parameters.

Our reply) Removed the methodology section in the introduction and added some information about the BDD film.

After) line 32-40: we added “Furthermore, research on boron-doped diamond (BDD), which imparts boron to diamond based on excellent diamond properties, gives diamonds electrical properties [6, 7]. Therefore many studies have been conducted on BDD because, BDD electrodes generate more hydroxyl radicals, more powerful oxidants, on the surface than the most widely used IrO2 electrodes. This results in excellent wastewater treatment [4, 5, 8, 9]. Because of these advantages, in the field of insoluble electrodes, much research is being conducted on BDD electrodes. However, despite these advantages, BDD is not widely used industrially because the production cost and price are too high to allow it to be used commercially as an electrode”.

  1. Why Nb interlayer was used and why columnar structure is so important?

Our reply) Nb interlayer was used because it is most widely used as a substrate for BDD electrodes.

Nb interlayer with columnar structure can prevent the phenomenon of bending, so that can prevent the diamond delamination after deposition

Additional comments have been added to line 60-66 for clarity.

Before) To solve this problem, PVD sputtering was used to coat the niobium interlayer with a columnar structure as shown in Figure 1b) before diamond and BDD deposition. This method, with a niobium interlayer, prevented the titanium substrate from bending as shown on the right in Figure 1a). Sputtering was performed through high-power impulse magnetron sputtering (HiPIMs), and the experimental conditions are shown in Table 1.

After) To solve this problem, PVD sputtering was used to coat the niobium, which is most widely used as a substrate for BDD electrodes, interlayer with a columnar structure as shown in Figure 1b) before diamond and BDD deposition. When returned to room temperature after deposition, the columnar structure of the niobium interlayer acts as a buffer between the two materials with different coefficients of thermal expansion. This method, with a niobium interlayer, prevented the titanium substrate from bending as shown on the right in Figure 1a). In addition, preventing the phenomenon of bending can prevent the diamond delamination after deposition. Sputtering was performed through high-power impulse magnetron sputtering (HiPIMs), and the experimental conditions are shown in Table 1. A columnar structure of niobium with a thickness of 3 μm was deposited by 1-hour deposition at 100 ℃.

  1. What was the wavelength of XRD source ?

Our reply) we added in line 123-124: The 2-theta scan was in the range of 30° to 80°. Cu Kα with the wavelength of λ = 1.5406 Å was used, and the acceleration voltage and current were 40 kV and 40 mA, respectively.

Before) XRD measurement conditions were measured in coupled scanning diffraction (CSD) mode measuring theta / 2theta. The 2 theta scan range was from 30 ° to 80 °, and the acceleration voltage and current were 40 kV and 40 mA, respectively.

After) XRD measurement conditions were measured in coupled scanning diffraction (CSD) mode measuring theta / 2theta. The 2-theta scan was in the range of 30° to 80°. Cu Kα with the wavelength of λ = 1.5406 Å was used, and the acceleration voltage and current were 40 kV and 40 mA, respectively.

  1. What was the size of exposed area during the electrochemical tests?

Our reply) The exposed area during the electrochemical test is 10 mm in diameter.

Before) Line 127-128: A potentiostat (WonATech Co, ZIVE SP2) was used to analyze the electrochemical properties of the BDD specimens by CV curve measurement. The potential window of oxygen and hydrogen reaction was measured using a 0.5-M Na2SO4 solution as an electrolyte, and the electrochemical activation and catalytic ability of BDD were measured by mixing a 0.5-M Na2SO4 solution and a 50-mM K3Fe(CN)6/K4Fe(CN)6 solution.

After) Line 127-128: A potentiostat (WonATech Co, ZIVE SP2) was used to analyze the electrochemical properties of the BDD specimens by CV curve measurement. The exposed area during the electrochemical test is 10 mm in diameter. The potential window of oxygen and hydrogen reaction was measured using a 0.5-M Na2SO4 solution as an electrolyte, and the electrochemical activation and catalytic ability of BDD were measured by mixing a 0.5-M Na2SO4 solution and a 50-mM K3Fe(CN)6/K4Fe(CN)6 solution.

  1. Line 133 What Authors mean writing “particle size”?

Our reply) It is particle size of non-doped diamond and reference BDD

Before) In terms of average particle size, the deposited diamond film is smaller than the reference BDD, but triangular (111) facets with superior diamond quality dominate.

After) In terms of average particle size, the non-doped diamond film is smaller than the reference BDD, but triangular (111) facets with superior diamond quality dominate.

  1. Why Authors did not compare Raman spectra of Diamond and BDD? The visible D and G bands may give a lot of information regarding the amorphous carbon concentration and quality.

Our reply) The current Raman spectra is the diamond before doping boron, not the Raman spectra of boron-doped diamonds. In response to your comments, we have added Raman spectra of boron-doped diamonds.

Due to the addition of the BDD Raman spectra, the original Figure 5 has been modified to Figure 4 and Raman has been replaced with the original 4 to 5 for clarity. There was also the movement and correction of the text due to the change of Figure order.

Before) Raman spectroscopy was conducted to analyze the quality of the deposited diamond, and the results are shown in Figure 4. Single-crystal diamond (SCD) was used as a reference for comparison with the deposited diamond. The diamond characterization peak is very distinctly observed at 1332 cm-1. In addition, D- and G-mode bands of the amorphous carbon region were observed at 1400-1650 cm-1 [25, 26]. Raman spectroscopy confirmed that a diamond film was successfully deposited.

After) Raman spectroscopy was conducted to analyze the quality of the non-doped diamond, and the results are shown in Figure 5a). Single-crystal diamond (SCD) was used as a reference for comparison with the non-doped diamond. The diamond characterization peak is very distinctly observed at 1332 cm-1. In addition, D- and G-mode bands of the amorphous carbon region were observed at 1400-1650 cm-1 [25, 26]. Raman spectroscopy confirmed that a diamond film was successfully deposited.

Raman analysis showed significant changes as the boron was doped, and there was no difference between the 12 hours and 60 hours. Raman spectra of BDD are quite different from those of non-doped diamond films. The corresponding diamond characterization peak at 1332 cm-1 in Figure 5b) and c) are absent. Instead, two broad peaks are observed, shifted to a lower frequency. Boron doping does not significantly affect the FESEM morphology of the diamond film, but it does have a large impact on the Raman spectrum.

Regarding this mismatch between the FESEM image and the Raman spectra, Ushizawa and Mortet found that when boron doping was higher than 400 ppm, the characteristic peak of the diamond changed. The diamond characterization peak gradually decreases, shifting to a lower frequency, and changes to a broader peak. This would refer to broad peaks found at around 1210 cm-1 in Figure 5b) and c). This is manifested by impurities and defects as a result of relaxation of wave vector selection rules due to disorder. Ushizawa and Mortet also note that when boron is doped with diamond, a broad peak at 490 cm-1 is also observed due to interference between continuous electron excitation and the center optic phonon in the discontinuous region. Thus, the Raman spectrum in Figure 5b) and c) indicates that boron was successfully doped into the diamond film.

  1. Why Authors do not present the XRD spectra of diamond film. Coatings on which substrates were analyzed by XRD?

Our reply) It was titanium substrate with niobium interlayer. The manuscript has been modified.

Before) The XRD pattern analysis results of the BDD films are shown in Figure 6.

After) The XRD pattern analysis results of the BDD films, that on titanium substrate with niobium interlayer, are shown in Figure 6.

  1. XRD results can clearly confirm the statement about a dominance of <111> and <100> type crystallographic planes.

Our reply) The overall modification of Figure 6 has been made. Indexing has been added for each peak. Please check Figure 6 in our revised manuscript.

  1. Figure 6. First should be presented a) and then b).

Our reply) The overall modification of Figure 6 has been made. Positions of a) and b) have been changed to make them clearer. Please check Figure 6 in our revised manuscript.

  1. The conclusions are not supported by the obtained results. Authors do not refer to any results from other papers. The discussion of the obtained results is at very poor scientific level. Authors rather state than explain the differences, whereas those statements are not supported by proper references.

Our reply) We carefully read your comments and revised conclusion in our manuscript.

After) The Raman spectra of the BDD films confirmed that boron was successfully doped into the diamond. However, there was no change in the morphology of diamond film due to boron doping. And the experimental results according to the deposition time, the deposition rate according to the increase in deposition time was confirmed to be constant at an average of 100 nm/h. Furthermore, through the evaluation of electrochemical properties, the BDD film samples had electrochemical properties that the diamond film did not have. As the thickness increased, there was an increase in the potential window, but there was no change in electrochemical activation and catalytic activity. This means that electrochemical activation and catalytic activity are only affected by the properties of the exposed surface, regardless of the thickness or density of the BDD film. Meanwhile, comparison with the reference BDD where amorphous carbonization occurred showed that increased thickness does not necessarily increase electrochemical properties. In other words, although a high deposition rate is an important factor in the deposition of BDD film, deposition under conditions in which amorphous carbonization does not occur due to temperature control during deposition can improve the electrochemical properties of BDD.

Round 2

Reviewer 1 Report

After revision, the article is acceptable.

Author Response

Thank you very much!